

# A glycolysis-related gene pairs signature predicts prognosis in patients with hepatocellular carcinoma

Weige Zhou[1], Shijing Zhang[1], Zheyou Cai[1], Fei Gao[2], Wenhui Deng[3], Yi Wen[4], Zhen-wen Qiu[4], Zheng-kun Hou[4] and Xin-Lin Chen[1]

[1] School of Basic Medical Science, Guangzhou University of Chinese Medicine, Guangzhou, China
[2] Department of Minimally Invasive & Interventional Radiology, Sun Yat-sen University Cancer Center, Guangzhou, China
[3] The Fourth Affiliated Hospital of Guangzhou University of Chinese Medicine, Shenzhen, China
[4] The First Affiliated Hospital of Guangzhou University of Chinese Medicine, Guangzhou, China

## ABSTRACT

**Background:** Hepatocellular carcinoma (HCC) is one of the most universal malignant liver tumors worldwide. However, there were no systematic studies to establish glycolysis-related gene pairs (GRGPs) signatures for the patients with HCC. Therefore, the study aimed to establish novel GRGPs signatures to better predict the prognosis of HCC.

**Methods:** Based on the data from Gene Expression Omnibus, The Cancer Genome Atlas (TCGA) and International Cancer Genome Consortium databases, glycolysis-related mRNAs were used to construct GRGPs. Cox regression was applied to establish a seventeen GRGPs signature in TCGA dataset, which was verified in two validation (European and American, and Asian) datasets.

**Results:** Seventeen prognostic GRGPs (HMMR_PFKFB1, CHST1_GYS2, MERTK_GYS2, GPC1_GYS2, LDHA_GOT2, IDUA_GNPDA1, IDUA_ME2, IDUA_G6PD, IDUA_GPC1, MPI_GPC1, SDC2_LDHA, PRPS1_PLOD2, GALK1_IER3, MET_PLOD2, GUSB_IGFBP3, IL13RA1_IGFBP3 and CYB5A_IGFBP3) were identified to be significantly progressive factors for the patients with HCC in the TCGA dataset, which constituted a GRGPs signature. The patients with HCC were classified into low-risk group and high-risk group based on the GRGPs signature. The GRGPs signature was a significantly independent prognostic indicator for the patients with HCC in TCGA (log-rank $P = 2.898e{-}14$). Consistent with the TCGA dataset, the patients in low-risk group had a longer OS in two validation datasets (European and American: $P = 1.143e{-}02$, and Asian: $P = 6.342e{-}08$). Additionally, the GRGPs signature was also validated as a significantly independent prognostic indicator in two validation datasets.

**Conclusion:** The seventeen GRGPs and their signature might be molecular biomarkers and therapeutic targets for the patients with HCC.

Corresponding author
Xin-Lin Chen, chenxlsums@126.com

## INTRODUCTION

Hepatocellular carcinoma (HCC), which is the second most dominant cause of cancer deaths throughout the word, is the most ordinary form of primary carcinoma of the liver (*Llovet et al., 2016*). It is estimated that approximately 841,000 new cases are expected to occur worldwide and more than 780,000 patients would die of HCC in 2018 (*Bray et al., 2018*). Kinase and immune checkpoint inhibitors have been shown to be effective options for the treatment of advanced-stage HCC, but they have limited effectiveness (*Yang et al., 2019b*). Despite the new progress in drug development, the clinical outcomes in patients with advanced HCC remains poor. Based on the Surveillance, Epidemiology, and End Results database, the 5-year survival rate was 30.5% for patients with local HCC, and less than 5% for those with distant metastasis (*Oweira et al., 2017*). Due to the poor outcomes, it is necessary to investigate novel effective markers for the prognosis of HCC.

Recent developments in high-throughput sequencing, technologies and bioinformatics have drastically changed research on genomic in disease, and many marker changes related to prognosis and survival have been revealed through mining databases (*Liu et al., 2018*). Several biomarkers have been shown to predict the prognosis of the patients with HCC. For example, serum lncRNA urothelial carcinoma-associated 1 (UCA1) was an independent harmful prognostic indicator for HCC (*Zheng et al., 2018*). Collagen triple helix repeat containing 1 (CTHRC1) may serve as a prognostic biomarker for HCC (*Zhou et al., 2019*). Serum acetylcarnitine is a meaningful biomarker reflecting HCC diagnosis and progression (*Lu et al., 2016*). Notably, a six glycolysis-related gene signature was found to predict survival in patients with HCC (*Jiang et al., 2019*). However, in view of the intrinsic biological heterogeneity of tumors and batch effects caused by different sequencing platforms, previous prognostic gene signature had to standardize gene expression profiles, which was very difficult for data processing. Moreover, compared with a single gene marker, multigene prognostic signatures are better alternatives for predicting prognosis and survival (*Chen et al., 2018*). Thus, a novel method which omits data standardization and scaling based on the relative ranking of gene expression levels has been used in this study. Many reliable results have been obtained in various studies by using this method (*Heinaniemi et al., 2013*; *Li et al., 2017*; *Popovici et al., 2012*).

Glycolysis, one of the most ancient metabolic processes, is a low-energy-providing pathway. The metabolic properties of cancer cells differed from those of normal cells (*Annibaldi & Widmann, 2010*). Cancer cells had rapid metabolic features which increased uptake of glucose and glycolysis (*Akram, 2013*). This allowed cancer cells to preferentially metabolize glucose through aerobic glycolysis, offering them with a progression advantage (*Hua et al., 2018*). Some studies indicated that aerobic glycolysis phenotype was associated with poor prognosis of HCC (*Cui et al., 2018*; *Guo et al., 2015*; *Hua et al., 2018*; *Lin et al., 2018*; *Xu et al., 2017*). However, there were no systematic studies to establish glycolysis-related gene pairs (GRGPs) signatures to predict the survival of patients with HCC. Therefore, it was necessary to establish multigene prognostic signatures for HCC using glycolysis-related genes pairs.

The purpose of this study was to construct GRGPs signatures to predict the prognosis in patients with HCC.

# MATERIALS AND METHODS

## Data source

The expression profiles and clinicopathological data of HCC and normal tissues were obtained from The Cancer Genome Atlas (TCGA, https://portal.gdc.cancer.gov/) and International Cancer Genome Consortium Japan (ICGC, https://dcc.icgc.org/releases/current/Projects/LIRI-JP). The clinical information inclusion criteria were set as follows: (1) patients had completely detailed clinical information; (2) The follow-up time of samples exceeded 30 days. GSE10140, GSE10141, GSE10143, GSE15654, GSE14520, GSE76427, GSE45114 expression profile was derived from the Gene Expression Omnibus (GEO) database, including 1,469 samples. The TCGA was used as training dataset. Other databases were used as validation datasets.

## Gene set enrichment analysis

To explore whether the specific gene sets were significant different between tumor group and normal group, we performed gene sets enrichment analysis (GSEA) (http://www.broadinstitute.org/gsea/index.jsp). The mRNAs expression levels between tumor and non-tumor groups were analyzed to confirm which function could be available for further study. Normalized $P$-value $\leq 0.05$ were considered to be statistically significant.

## Construction and evaluation of glycolysis-related gene pairs signature

First of all, the glycolysis-related mRNAs level in the same sample was pairwise compared to generate a score for each glycolysis-related gene pair (GRGP). If the expression level of gene 1 was greater than gene 2, the GRGP score was 1, otherwise it was 0 (Li et al., 2017). GRGPs with high variation in TCGA dataset (median absolute deviation >0.05) were included in subsequent model construction. The prognostic value of GRGPs was identified by univariate Cox regression. Then, GRGPs with $P \leq 0.05$ in univariate analysis were incorporated into Lasso regression model in order to establish a GRGPs signature. A risk score was established according to the following formula: risk score $= \sum_i \text{Coefficient} (\text{GRGP}_i) * \text{Score} (\text{GRGP}_i)$. To classify patients into low-risk group and high-risk group, the optimal cut off of the GRGPs signature was defined using time-dependent receiver operating characteristic (ROC) curve analysis at 1 year in the TCGA dataset for overall survival (OS). The OS between two groups was compared utilizing Kaplan–Meier and Log-rank test. Risk score and other clinicopathological characteristics were included in the model so as to confirm whether risk score was an independent factor to predict the progress of the patients. Further, the clinical value of the GRGPs signature was verified by comparing the risk scores of patients with different ages, gender, grade and stage.

The OS of the patients with HCC at 1 year, 3 years and 5 years were predicted using a nomogram. Index of concordance (C-index) and Calibration curves were applied to explore the accuracy of the model established from TCGA dataset.

## Validation of the GRGPs signature

Seven GEO databases (GSE10140, GSE10141, GSE10143, GSE15654, GSE14520, GSE76427, and GSE45114) and ICGC database were enrolled for subsequent verification. According to human race, all the databases were divided into European and American dataset and Asian dataset. European and American dataset (765 samples) included GSE10140, GSE10141, GSE10143, and GSE15654 database; Asian dataset (947 samples) included GSE14520, GSE76427, GSE45114 and ICGC database. Due to the lack of detailed clinical information (such as age, gender, stage) in European and American dataset, thus it was only used for survival analysis. Asian dataset was used for subsequent survival analysis and independent prognostic analysis. The GRGPs signature was calculated, and then Kaplan–Meier, Log-rank test and Cox regression were used to verify whether the GRGPs signature was significantly associated with OS. The ROC curve, C-index and Calibration curves were constructed to examine whether the model established by TCGA dataset could effectively predict survival in patients with HCC.

## Statistical analysis

Cox regression was utilized to evaluate the associations between the expression levels of the glycolytic-related mRNAs and OS. Moreover, univariate and multivariate Cox regression were applied to determine the prognostic values of the GRGPs signature and various clinicopathological characteristics. The prediction accuracy of the risk score for 1-year, 3-years and 5-years survival was evaluated using ROC curve analysis. Statistical tests were two-sided, and $P$ values below 0.05 were considered to be statistically significant.

# RESULTS

## Preliminary selection of genes using GSEA

According to inclusion criteria, 349 patients with HCC were obtained. Expression signatures of marker gene sets were obtained by condensing multiple gene sets from the Molecular Signature Database (MSigDB). Each expression signature involved 50 specific gene sets that stand for clearly defined biological statuses and processes (*Zhang, Zhang & Yu, 2019*). GSEA was applied to investigate whether the identified gene sets revealed significant differences between the tumor and normal groups. Forty-four gene sets were upregulated in HCC, and 20 gene sets were highly enriched, with normalized $P < 0.05$ among the 50 gene sets (Fig. 1; Table S1). As can be seen in the Table S1, the biological process of glycolysis ($P < 0.05$) contained 199 genes, which was the second largest in size.

## Identification of prognostic GRGPs

Based on 199 GRGs, 19,701 GRGPs were established. After removing GRGPs with comparatively low variation (MAD > 0.05), 1,102 GRGPs were reserved and selected as initial candidate GRGPs. Cox regression was utilized to select prognostic GRGPs. According to the results of univariate Cox, 35 GRGPs had prognostic values for the patients with HCC ($P < 0.05$, Table S2). Subsequently, lasso regression model revealed that 17 GRGPs as prognostic factors (Table 1; Figs. 2A–2B). As shown in Table 1, five GRGPs (HMMR_PFKFB1, CHST1_GYS2, MERTK_GYS2, GPC1_GYS2 and LDHA_GOT2) were

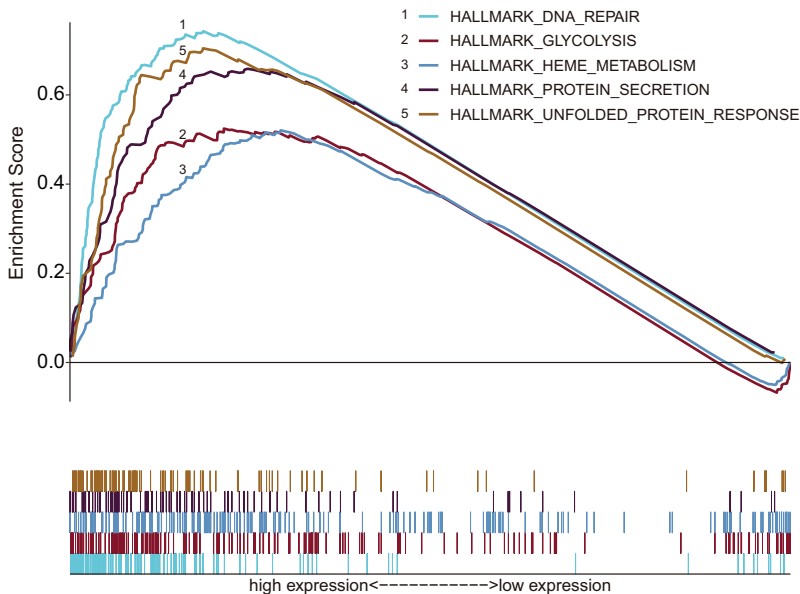

**Figure 1 Enrichment graph of five gene sets with significant differences between tumor and non-tumor tissues.** The high expression of these genes was principally enriched in biological processes such as glycolysis, DNA repair, metabolism and protein synthesis secretion.

found to be harmful prognostic factors and twelve GRGPs (IDUA_GNPDA1, IDUA_ME2, IDUA_G6PD, IDUA_GPC1, MPI_GPC1, SDC2_LDHA, PRPS1_PLOD2, GALK1_IER3, MET_PLOD2, GUSB_IGFBP3, IL13RA1_IGFBP3 and CYB5A_IGFBP3) were verified to be beneficial prognostic factors.

## Construction of GRGPs signature

These seventeen GRGPs were utilized to establish a GRGPs signature. Risk score of GRGPs signature for each patient was calculated utilizing the following formula (formula 1): risk score = $(-0.30794 * IDUA\_GNPDA1) - (0.15299 * IDUA\_ME2) - (0.16389 * IDUA\_G6PD) - (0.35599 * IDUA\_GPC1) + (0.04846 * HMMR\_PFKFB1) - (0.35632 * MPI\_GPC1) - (0.29752 * SDC2\_LDHA) - (0.09077 * PRPS1\_PLOD2) - (0.06137 * GALK1\_IER3) + (0.02511 * CHST1\_GYS2) - (0.26287 * MET\_PLOD2) - (0.00305 * GUSB\_IGFBP3) + (0.34302 * MERTK\_GYS2) + (0.20608 * GPC1\_GYS2) - (0.31484 * IL13RA1\_IGFBP3) + (0.17629 * LDHA\_GOT2) - (0.10962 * CYB5A\_IGFBP3)$. The cutoff point of risk score was set at −0.698 utilizing ROC curve analysis, which classified the patients into high-risk group or low-risk group (Fig. 2C). Risk score was significantly associated with OS of the patients with HCC, in which OS in low-risk group was longer than that in high-risk group ($P = 2.898e-14$, Fig. 3A). The survival time of patients with HCC decreased with risk score increasing (Fig. 4).

## Association between risk score and clinicopathologic factors

Risk score increased with age, stage, and survival status, demonstrating that the GRGPs signature might be relevant to the progression of HCC. Risk score of patients with

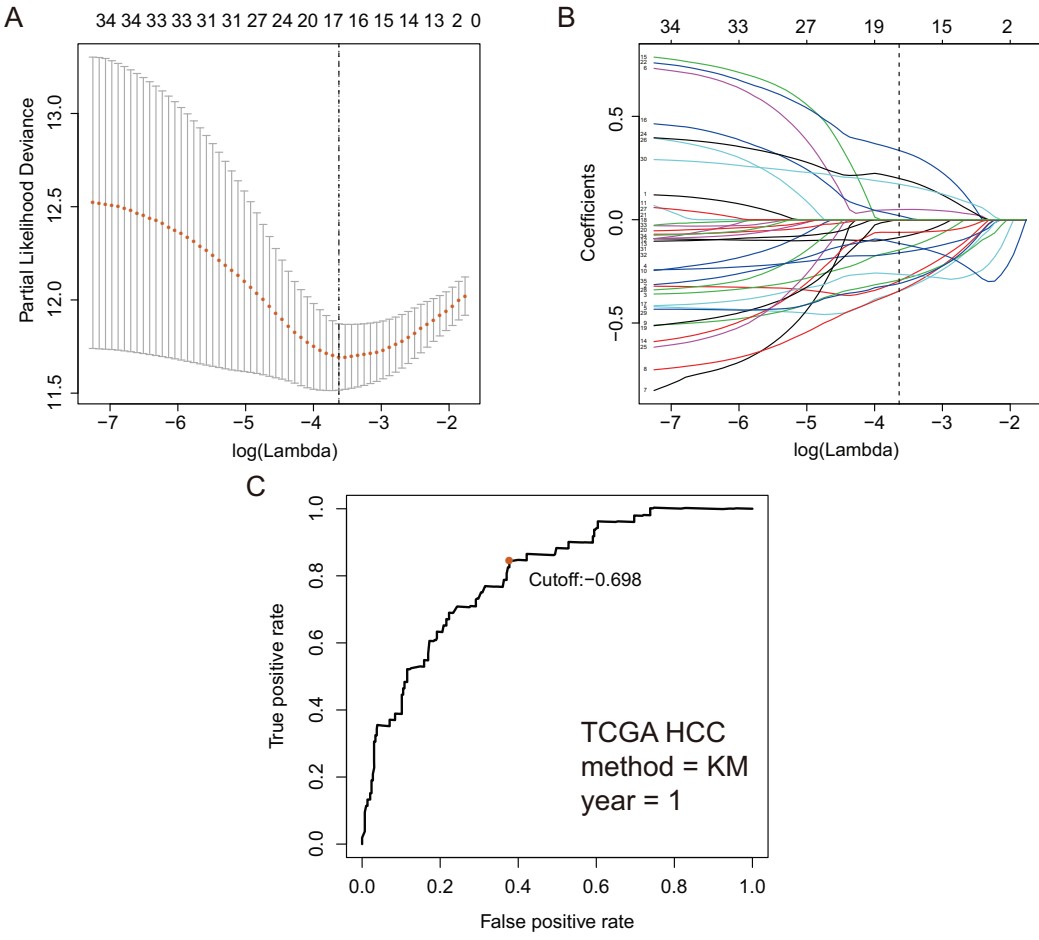

**Figure 2 GRGPs selection utilizing lasso model based on TCGA dataset.** (A) Elastic net regularization course with partial likelihood deviance plot. The vertical dashed line with minimum partial likelihood deviance value is at the optimal logarithmic (Lambda) value. Lambda is the parameter which controls the regulation degree of lasso regression complexity. The ordinate is the value of the coefficient, the lower abscissa is log (lambda), and the upper abscissa is the number of non-zero coefficients in the model. (B) Lasso coefficient values of 17 prognosis GRGPs. Each colored line represents the change track of each independent variable coefficient. (C) Time-dependent ROC curve for GRGPs at 1 year. GRGPs score of −0.698 was utilized as cutoff point for GRGPs.

advanced-stage and advanced-age were significantly higher than those with early-stage and early-age (Figs. S1A and S1B). Risk scores of dead patients were higher than those of living patients (Fig. S1C). The heat map illustrated that the high expression of these seventeen GRGPs were significantly related to female, lower survival status of patients, higher stage and higher grade (Fig. S1D).

## The GRGPs signature was an independent prognostic factor

In TCGA dataset, univariate Cox regression revealed that GRGPs signature was associated with OS and its HR was 3.508 (95% CI [2.608–4.720], $P < 0.001$, Table S3; Fig. 5A). After controlling clinical features such as gender, age, tumor stage and grade, GRGPs signature remained to be an independent prognostic indicator (HR = 3.204, 95% CI [2.293–4.476], $P < 0.001$, Table 2; Fig. 5B). GRGPs signature and TNM stage were

**Table 1 Lasso regression coefficients and molecular function information of seventeen GRGPs based on TCGA-HCC data.**

| Gene1 | Encoding protein | Function | Gene2 | Encoding protein | Function | Coefficient |
|---|---|---|---|---|---|---|
| IDUA | hydrolyzes the terminal alpha-L-iduronic acid residues of two glycosaminoglycans, dermatan sulfate and heparan sulfate | Chondroitin sulfate/dermatan sulfate metabolism and Glycosaminoglycan metabolism | GNPDA1 | An allosteric enzyme | The reversible conversion of D-glucosamine-6-phosphate into D-fructose-6-phosphate and ammonium | −0.294 |
| IDUA | hydrolyzes the terminal alpha-L-iduronic acid residues of two glycosaminoglycans, dermatan sulfate and heparan sulfate | Chondroitin sulfate/dermatan sulfate metabolism and Glycosaminoglycan metabolism | ME2 | A mitochondrial NAD-dependent malic enzyme | Catalyzes the oxidative decarboxylation of malate to pyruvate | −0.145 |
| IDUA | hydrolyzes the terminal alpha-L-iduronic acid residues of two glycosaminoglycans, dermatan sulfate and heparan sulfate | Chondroitin sulfate/dermatan sulfate metabolism and Glycosaminoglycan metabolism | G6PD | A cytosolic enzyme encoded by a housekeeping X-linked gene | Produce NADPH | −0.160 |
| IDUA | hydrolyzes the terminal alpha-L-iduronic acid residues of two glycosaminoglycans, dermatan sulfate and heparan sulfate | Chondroitin sulfate/dermatan sulfate metabolism and Glycosaminoglycan metabolism | GPC1 | Disease related genes belongs to the glypican family | Play a role in the control of cell division and growth regulation | −0.345 |
| HMMR | Involved in cell motility | Regulation of PLK1 activity at G2/M transition and metabolism | PFKFB1 | A member of the family of bifunctional 6-phosphofructo-2-kinase | An activator of the glycolysis pathway and an inhibitor of the gluconeogenesis pathway/participate in hepatocellular carcinoma tumorigenesis | 0.049 |
| MPI | Phosphomannose isomerase catalyzes the interconversion of fructose-6-phosphate and mannose-6-phosphate | Metabolism of proteins and amino sugar and nucleotide sugar metabolism | GPC1 | Disease related genes belongs to the glypican family | Play a role in the control of cell division and growth regulation | −0.343 |
| SDC2 | A transmembrane (type I) heparan sulfate proteoglycan and is a member of the syndecan proteoglycan family | Microglia activation during neuroinflammation: overview and cell surface interactions at the vascular wall | LDHA | Cancer-related protein belongs to the LDH/MDH superfamily | Catalyzes the conversion of L-lactate and NAD to pyruvate and NADH in the final step of anaerobic glycolysis | −0.291 |
| PRPS1 | Catalyzes the phosphoribosylation of ribose 5-phosphate to 5-phosphoribosyl-1-pyrophosphate | Thiopurine pathway, pharmacokinetics/pharmacodynamics and carbon metabolism | PLOD2 | A membrane-bound homodimeric enzyme | Participate in collagen chain trimerization and degradation of the extracellular matrix | −0.085 |
| GALK1 | Galactokinase is a major enzyme for the metabolism of galactose | Galactokinase is a major enzyme for the metabolism of galactose | IER3 | A predicted intracellular protein belongs to the IER3 family | Protect cells from Fas- or tumor necrosis factor type alpha-induced apoptosis | −0.061 |
| CHST1 | A member of the keratin sulfotransferase family of proteins. The encoded enzyme catalyzes the sulfation of the proteoglycan keratin | Among its related pathways are Keratan sulfate/keratin metabolism and metabolism | GYS2 | Liver glycogen synthase | Participate in galactose metabolism and glycogen metabolism | 0.019 |

(Continued)

| Gene1 | Encoding protein | Function | Gene2 | Encoding protein | Function | Coefficient |
|---|---|---|---|---|---|---|
| MET | A member of the receptor tyrosine kinase family of proteins and the product of the proto-oncogene MET | Hepatocyte growth factor, induces dimerization and activation of the receptor | PLOD2 | A membrane-bound homodimeric enzyme | Participate in collagen chain trimerization and degradation of the extracellular matrix | −0.265 |
| MERTK | A member of the MER/AXL/TYRO3 receptor kinase family | Regulate cell survival, migration, differentiation, and phagocytosis of apoptotic cells | GYS2 | Liver glycogen synthase | Participate in galactose metabolism and glycogen metabolism | 0.334 |
| GPC1 | Disease related genes belongs to the glypican family | Play a role in the control of cell division and growth regulation | GYS2 | Liver glycogen synthase | Participate in galactose metabolism and glycogen metabolism | 0.198 |
| IL13RA1 | A subunit of the interleukin 13 receptor | Bind tyrosine kinase TYK2 and mediate the signaling processes | IGFBP3 | Encodes a protein with an IGFBP domain and a thyroglobulin type-I domain | Prolonging the half-life of insulin-like growth factor (IGF) and altering their interaction with cell surface receptors | −0.308 |
| LDHA | Cancer-related protein belongs to the LDH/MDH superfamily | Catalyzes the conversion of L-lactate and NAD to pyruvate and NADH in the final step of anaerobic glycolysis | GOT2 | A pyridoxal phosphate-dependent enzyme | Play a role in amino acid metabolism and the urea and tricarboxylic acid cycles. | 0.171 |
| CYB5A | A membrane-bound cytochrome | Reduces ferric hemoglobin (methemoglobin) to ferrous hemoglobin | IGFBP3 | Encodes a protein with an IGFBP domain and a thyroglobulin type-I domain | Prolonging the half-life of insulin-like growth factor (IGF) and altering their interaction with cell surface receptors | −0.116 |

Note:
PLK1, Polo-like kinase 1; MET, Mesenchymal Epithelial Transition; MER/AXL/TYRO3 receptor, TAM receptors; TYK2, Tyrosine Kinase 2; LDH/MDH, lactate and Malate dehydrogenases; NAD, Nicotinamide adenine dinucleotide; NADH, Nicotinamide adenine dinucleotide; IGFBP, insulin-like growth factor-binding protein; IGF, insulin-like growth factor; IER3, Immediate Early Response 3; NADPH, nicotinamide adenine dinucleotide phosphate.

independent prognostic factors based on the TCGA. Thus, these factors were included in nomogram. GRGPs signature was the largest contributor to 1-year, 3-year and 5-year OS (Fig. 6A). The C-index of the nomogram was 0.764 (95% CI [0.715–0.813]). The areas under the ROC curve (AUC) corresponding to the survival at 1 year, 3 years and 5 years were 0.803, 0.777 and 0.774, respectively ($P < 0.05$). The C-index, ROC curve and Calibration explained that the GRGPs signature had better accuracy (Figs. 6B–6D).

## Validation of the GRGPs signature

In validation datasets, the risk score of GRGPs signature was calculated according to formula 1. The risk score was also significantly correlated with OS of patients with HCC (European and American dataset: $P = 1.143e−02$, Fig. 3B; Asian dataset: $P = 6.342e−08$, Fig. 3C). Univariate independent prognostic analysis indicated that GRGPs signature were independent prognostic factors in Asian dataset (HR of risk score = 2.661, 95% CI [1.862–3.803], $P < 0.001$, Table S4; Fig. 5C). After controlling stage and gender, GRGPs signature remained an independent prognostic indicator in multivariate analysis (HR = 2.567, 95% CI [1.714–3.844], $P < 0.001$, Table S5; Fig. 5D). According to results of

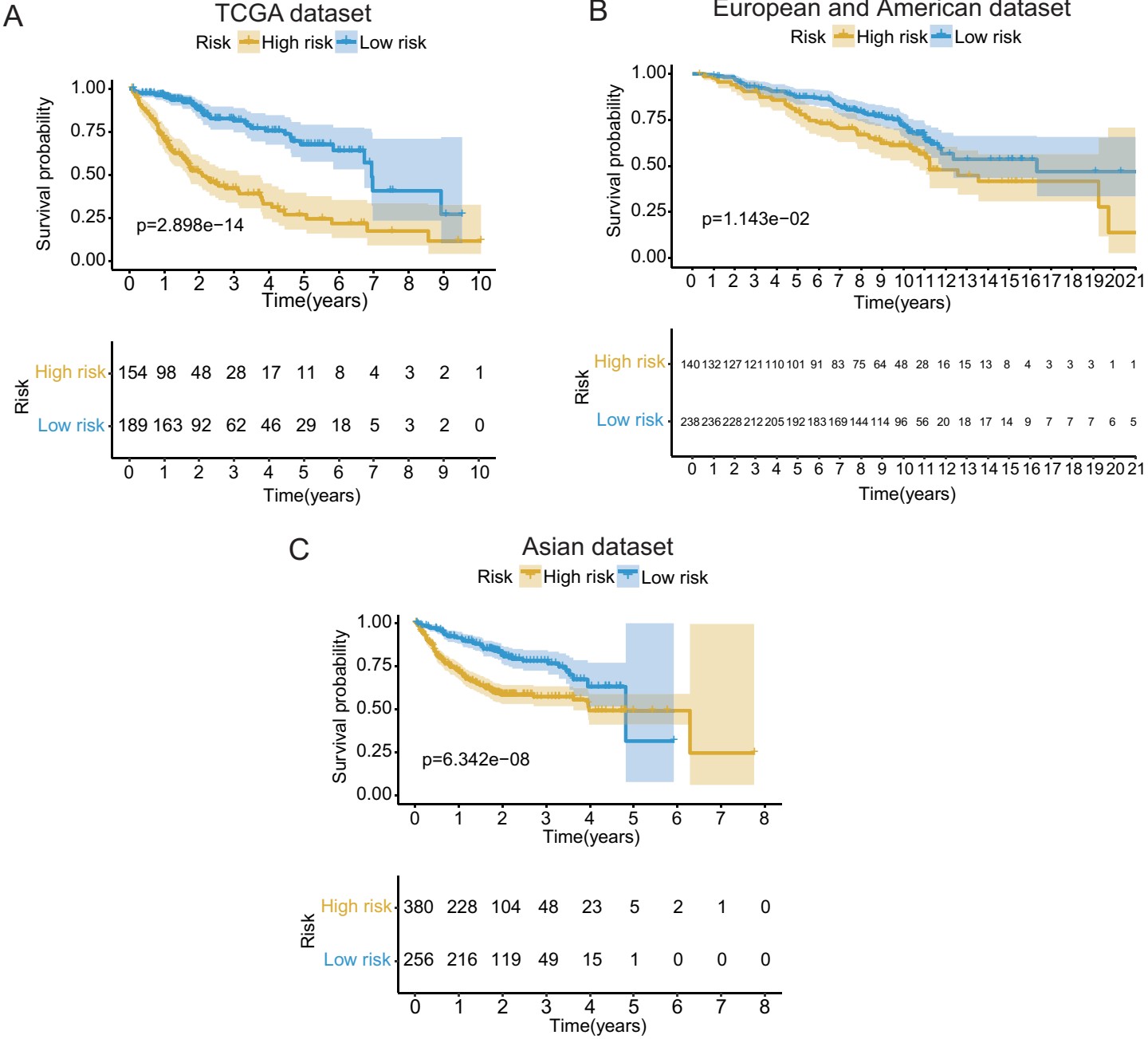

**Figure 3 The Kaplan–Meier (KM) survival curves of the GRGPs signature for patients with HCC based on TCGA and two validation datasets.** (A) The KM survival curves of TCGA dataset demonstrated that high-risk group had shorter OS period contrasted with low-risk group (*P* < 0.001). (B and C) Consistent with TCGA dataset, the OS of patients in high-risk group was shorter than that in low-risk group in two validation datasets (*P* < 0.05).

independent prognostic analysis, GRGPs signature, stage and gender were included in nomogram based on Asian dataset. GRGPs signature and age were the largest contributor to 1-year, 3-year and 5-year OS in Asian dataset nomogram (Fig. 6E). The C-index of nomogram based on Asian dataset was 0.705 (95% CI [0.660–0.750]). The areas under the

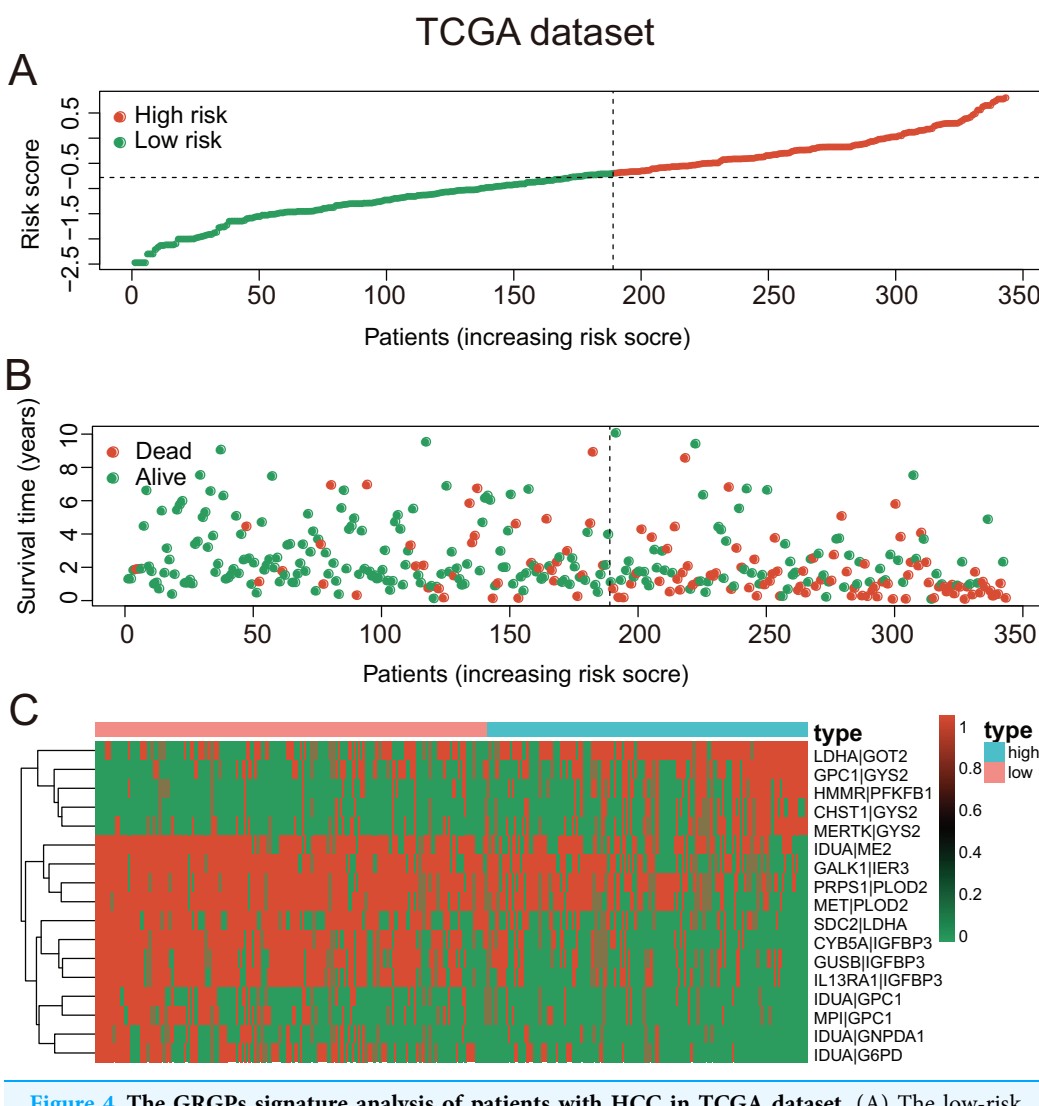

**Figure 4 The GRGPs signature analysis of patients with HCC in TCGA dataset.** (A) The low-risk group and high-risk group for the GRGPs signature in patients with HCC. (B) The survival status and time of patients with HCC. (C) Visualized heat map of the seventeen vital prognosis GRGPs expression in patients with HCC. The color from green to red reveals a rising tendency from low to high levels.

ROC curve at 1 year, 3 years and 5 years in Asian dataset were 0.694, 0.664, 0.536, respectively (Fig. 6H). The 3-year and 5-year calibration curves also proved that the GRGPs signature had great accuracy and robustness (Figs. 6F and 6G).

## DISCUSSION

HCC is one of the most universal malignant liver tumors worldwide. Long-term prognosis for HCC remains mighty poor, with metastasis being the major cause of mortality (*Uchino et al., 2011*). Most tumor cells support synthetic growth and evade apoptosis through aerobic glycolysis. (*Iansante et al., 2015*; *Warburg, 1956*). Glycolytic transcriptional factors and glycolysis-related proteins in cancers are significantly correlated with poor prognosis,

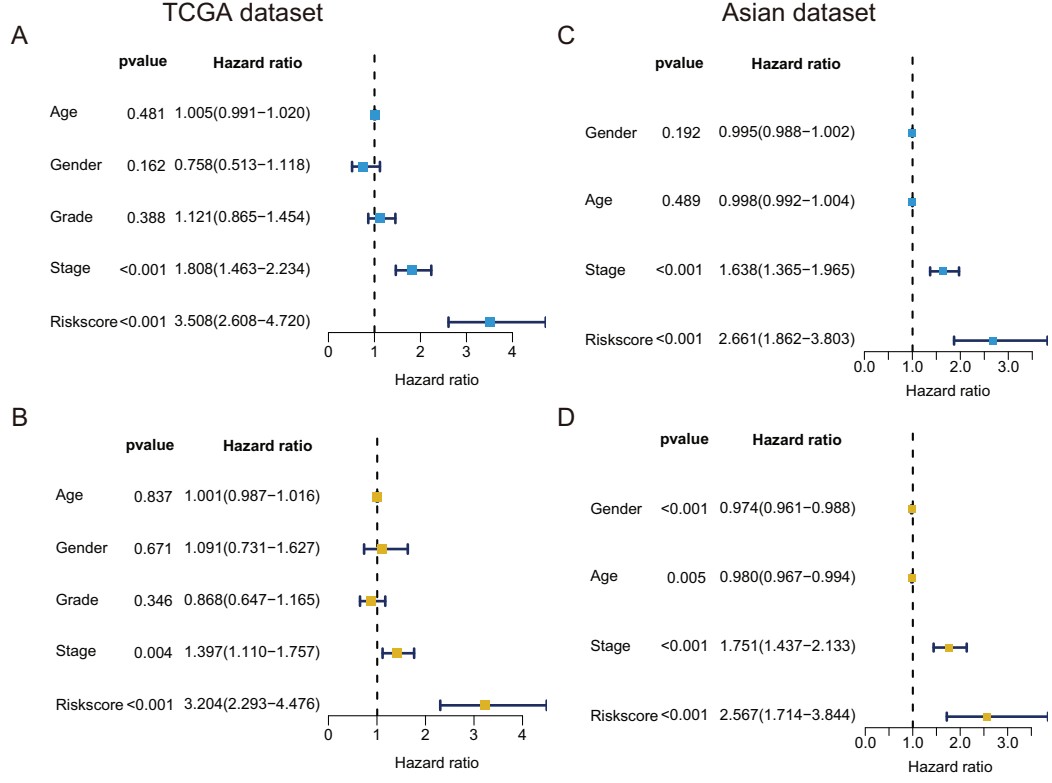

**Figure 5 Prognostic indicators based on GRGPs signature revealed great clinical values in TCGA dataset and Asian dataset.** Univariate (A) and multivariate (B) Cox regression of the relevancy between clinical features (containing the risk score) and OS of patients with HCC in the TCGA dataset. Univariate (C) and multivariate (D) Cox regression of the relevancy between clinical indicators (including the risk score) and OS of patients with HCC in the Asian dataset.

indicating that glycolytic status may be potentially valuable prognostic biomarkers for various cancers (*Yu et al., 2019*). Thus, it was valuable and urgent to establish a GRGPs signature in term of predicting the prognosis of patients with HCC.

In consideration of the intrinsic biological heterogeneity of tumors and batch effects caused by different sequencing platforms, previous gene signature needed to standardize or scale gene expression profiles, which resulted in the inability to process large amounts of data quickly. The construction method of prognostic model using gene pairs in this study can overcome these shortcomings greatly, which has been shown to have high stability and accuracy in some studies (*Heinaniemi et al., 2013*; *Li et al., 2017*; *Popovici et al., 2012*).

In this study, a GRGPs signature was constructed to predict the prognosis of patients with HCC, whose accuracy was better than gene signature previously constructed (*Jiang et al., 2019*). The HCC patients from TCGA dataset could be classified into low-risk group and high-risk group using the optimal cutoff point determined by ROC curve ($P = 2.898e{-}14$). The high-risk patients with HCC had shorter OS than the low-risk patients. Consistent with TCGA dataset, The OS in low-risk patients was longer than that in high-risk patients in two validation datasets. The AUC in TCGA and validation datasets

**Table 2 Clinical characteristics and risk score of HCC utilizing multivariate cox regression in the TCGA dataset.**

| Variable | B | SE | z | HR | HR.95L | HR.95H | P value |
|---|---|---|---|---|---|---|---|
| Age | 0.001 | 0.007 | 0.205 | 1.001 | 0.987 | 1.016 | 0.837 |
| Gender | 0.087 | 0.204 | 0.425 | 1.091 | 0.731 | 1.627 | 0.671 |
| Grade | −0.141 | 0.150 | −0.942 | 0.868 | 0.647 | 1.165 | 0.346 |
| Stage | 0.334 | 0.117 | 2.853 | 1.397 | 1.110 | 1.757 | 0.004 |
| Risk score | 1.164 | 0.171 | 6.824 | 3.204 | 2.293 | 4.476 | <0.001 |

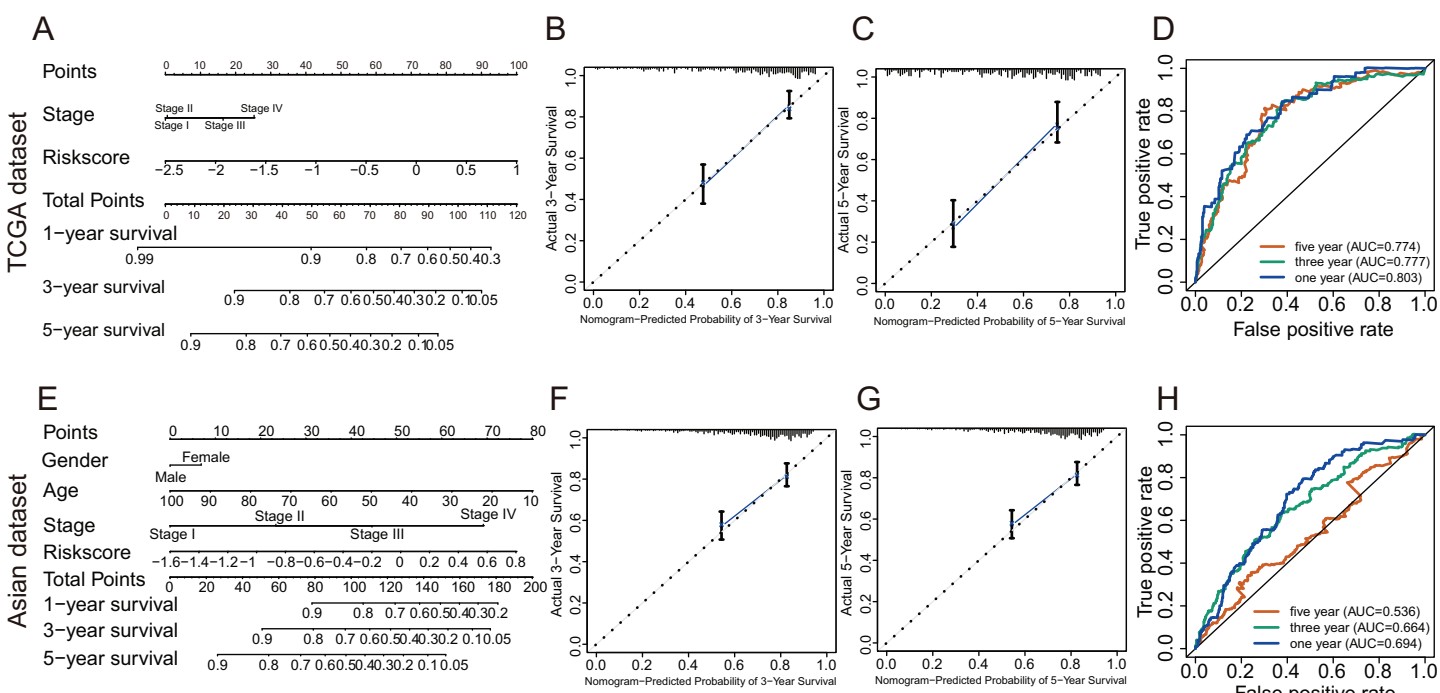

**Figure 6 The evaluation of prognostic GRGPs signature in the TCGA dataset and the Asian dataset.** (A and E) The nomogram figure about 1-year, 3-year or 5-year OS in HCC based on the TCGA dataset (A) or the Asian dataset (E). The point in the nomogram represented the individual score of each variable under different values. Total points represented the sum of the individual scores corresponding to all variables. For a single variable, we could get the corresponding point by drawing a vertical line upward, which must be perpendicular to the point line. For example, if someone's risk score is −1, the corresponding point of risk score in nomogram based on the TCGA dataset was about 42.5 by drawing a vertical line upward. Similarly, the corresponding point of the third stage was about 16.5. Then, add the points of all variables to get the total points of the patient (59). Based on the total point, the corresponding 1-year survival rate of the patient was about 0.86. (B, C, F and G) Calibration plots of 3-year (B) and 5-year (C) based on the TCGA dataset. Calibration plots of 3-year (F) and 5-year (G) based on the Asian dataset. Calibration plots for evaluating the agreement between the predicted and the actual OS for the model established by GRGPs. The 45° reference line indicates perfect calibration, where the predicted probabilities are consistent with the actual probabilities. (D and H) The areas under the ROC curve corresponding to 1 year, 3 years and 5 years of survival in the TCGA (D) or Asian datasets (H). The higher area under the ROC curve meant greater model accuracy.

were integrally greater than 0.6 which revealed that the GRGPs signature had certain accuracy in predicting survival. Both univariate and multivariate analysis indicated that the GRGPs signature could be set as an independent prognostic factor to predict the prognosis of patients with HCC in TCGA dataset and validation dataset. Furthermore, nomograms

were established based on the GRGPs signature and other clinical characteristics, which might serve as potential predictive tools for patients with HCC.

Seventeen GRGPs (HMMR_PFKFB1, CHST1_GYS2, MERTK_GYS2, GPC1_GYS2, LDHA_GOT2, IDUA_GNPDA1, IDUA_ME2, IDUA_G6PD, IDUA_GPC1, MPI_GPC1, SDC2_LDHA, PRPS1_PLOD2, GALK1_IER3, MET_PLOD2, GUSB_IGFBP3, IL13RA1_IGFBP3 and CYB5A_IGFBP3) were associated to OS of patients with HCC, which might be molecular markers of prognosis and potential therapeutic targets.

The prognostic signature consists of 17 GRGPs including 23 unique GRGs. Most of the GRGs involved in this signature are metabolism and tumor related genes that play an important role in patient prognosis and tumor metabolism. GYS is the rate-limiting enzyme for glycogen synthesis, which consists of two isoforms: GYS1 and GYS2 (*Roach et al., 2012*; *Szymanska et al., 2015*). Previous studies revealed that loss of GYS2 leaded to glycogen storage disease type 0 (GSD-0) with the symptom of weakened glucose tolerance (*Orho et al., 1998*; *Szymanska et al., 2015*). A recent study indicated that decreased expression levels of GYS2 reduced glycogen and significantly correlated with metastasis and poor prognosis of the patients with HCC, GYS2 restricted HBV-Related HCC growth through negative feedback loop with p53 (*Chen et al., 2019*). *Savci-Heijink et al. (2019)* demonstrated that the expression level of IDUA was down-regulated in patients with breast cancer and IDUA could be used as one of the potential targets for distinguishing whether breast cancer patients will undergo visceral metastasis (*Savci-Heijink et al., 2019*). A common growth factor co-receptor, Glypican-1 (GPC1), is abnormally rich in pancreatic cancer and that GPC1 deficiency inhibits tumor growth, angiogenesis and metastasis (*Aikawa et al., 2008*). Enhanced expression level of GPC1 is associated with BMP and activin receptors in pancreatic cancer, and the low expression of GPC1 could suppress pancreatic cancer cell growth (*Kayed et al., 2006*). It was found that increased expression level of GPC1 was significantly relevant with poor prognosis of the patients with pancreatic ductal adenocarcinoma (*Lu et al., 2017*; *Zhou et al., 2018*). The overexpression of GPC1 was correlated with poor prognosis of the patients with esophageal squamous cell carcinoma, and GPC1 is a key molecule that alters the threshold of chemoresistance to chemo-sensitivity against cis-Diammineplatinum (II) dichloride (CDDP) (*Hara et al., 2016*). However, there were no studies to report GPC1 prognostic role in HCC. Based on the results of this study, it was revealed to be potential molecular biomarkers and therapeutic targets for the HCC patients. Thus, more researches were necessary to figure out how GPC1 affects the prognosis of HCC exactly. In this study, HMMR was considered as potential molecular target for the treatment of HCC. HMMR has not been directly reported as therapeutic target for HCC. The Y-linked proto-oncogene could promote the expression of HMMR, which was correlated with poor prognosis in the patients with HCC (*Kido & Lau, 2019*). Over-expression of HMMR was verified as indicators of poor prognosis and metastasis in lung cancer (*Liu et al., 2019*; *Zhang, Zhang & Yu, 2019*). HMMR was confirmed to be a potential independent indicator of predicting survival in patients with papillary muscle-invasive bladder cancer (*Wang et al., 2019*; *Yang et al., 2019a*).

Several limitations exist in the current study. First, the study was a retrospective study, although we tried to incorporate as many datasets as possible to verify this signature more rigorously. More prospective studies was demanded to prove the prognostic functions of glycolysis-related signals. Second, Gene expression signatures are susceptible to sampling deviation caused by intratumor heterogeneity. Although we removed low variation GRGPs so as to reduce cross-study batch effects, some may still reserve genetic heterogeneity. Third, the functional experiments should be conducted to further indicate the potential molecular mechanisms for predicting effects of GRGPs.

## CONCLUSION

Our study systematically demonstrated the expression of glycolysis-related mRNAs in HCC, verifying the prognostic value of these mRNAs. The GRGPs signature could predict survival in patients with HCC. Therefore, the seventeen GRGPs and their signature may be molecular biomarkers and therapeutic targets for the patients with HCC, which conduces to explore new treatments for HCC.

### Funding

This study was funded by the National Natural Science Foundation of China (81774451), the Natural Science Foundation of Guangdong Province (2017A030313827), and the Science Program for Overseas Scholar (Xinhuo plan) of Guangzhou University of Chinese Medicine. The funders had no role in study design, data collection and analysis, decision to publish, or preparation of the manuscript.

### Grant Disclosures

The following grant information was disclosed by the authors:
National Natural Science Foundation of China: 81774451.
Natural Science Foundation of Guangdong Province: 2017A030313827.
Guangzhou University of Chinese Medicine.

### Competing Interests

The authors declare that they have no competing interests.

### Author Contributions

- Weige Zhou conceived and designed the experiments, performed the experiments, analyzed the data, prepared figures and/or tables, authored or reviewed drafts of the paper, and approved the final draft.
- Shijing Zhang performed the experiments, analyzed the data, prepared figures and/or tables, authored or reviewed drafts of the paper, and approved the final draft.
- Zheyou Cai performed the experiments, analyzed the data, prepared figures and/or tables, authored or reviewed drafts of the paper, and approved the final draft.
- Fei Gao conceived and designed the experiments, authored or reviewed drafts of the paper, and approved the final draft.

- Wenhui Deng performed the experiments, analyzed the data, prepared figures and/or tables, authored or reviewed drafts of the paper, and approved the final draft.
- Yi Wen analyzed the data, prepared figures and/or tables, authored or reviewed drafts of the paper, and approved the final draft.
- Zhen-wen Qiu performed the experiments, authored or reviewed drafts of the paper, and approved the final draft.
- Zheng-kun Hou performed the experiments, authored or reviewed drafts of the paper, and approved the final draft.
- Xin-Lin Chen conceived and designed the experiments, authored or reviewed drafts of the paper, and approved the final draft.

## Data Availability

Data are available at NCBI GEO: GSE10140, GSE10141, GSE10143, GSE15654, GSE14520, GSE76427, GSE45114.

## Supplemental Information

Supplemental information for this article can be found online at http://dx.doi.org/10.7717/peerj.9944#supplemental-information.

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
