# Peer review of "A glycolysis-related gene pairs signature predicts prognosis in patients with hepatocellular carcinoma"

_PeerJ, doi:10.7717/peerj.9944_

## Round 0.1 · original submission · Minor Revisions

Please address critiques of all reviewers and revise the manuscript accordingly.

·

Basic reporting

The study wanted to establish novel GRGPs signatures to better diagnose and predict HCC. The authors used clear English to explan the problem of interest, statistical methods and results. Sufficient background in the HCC prediction field was provided. The figures and tables were also clearly provided.
However, there is one part in the figure that needs modification:
Figure 3B is missing a vertical line (mentioned in the caption but missing in the figire).

Experimental design

The authors clearly stated the gap in establishing GRGPs signatures for patients with HCC. They used statisticcal methods to establish the signatures in one cohort and validated that in another cohort. The selection of gene was also reasonable. The authors compared 26 glycolysis-related DEmRNA between HCC and normal tissues to construct the GRGP signature. However, there are two parts that is not quite clear:
1. Line 162 to 164 in the word document: "Forty-four gene sets were upregulated in HCC, and 20 gene sets were highly enriched, with normalized P < 0.05 among the 50 gene sets (Fig. 1, Table S1)". This part is a little bit confusing. It would be good if more explanation on the table and figure is provided
2. Line 175 to 176 in the word document: "k = 2 seemed to be a more appropriate clustering choice". It would be better if the authors could give one or two sentences explanation on why k=2 is the best choice.

Validity of the findings

The authors clearly showed the data in the figures and tables. They also gave reasonable tests on whether GRGPs signature is independent and whether it has better accuracy than TNM stage.
The authors got the conclusion using TCGA dataset, they also provided vaildation of the GRGPs signature using ICGC cohort, which is self-consistent.
Overall the result is reasonable. But it would be great if more details are provided on the discovery of the seven GPGRs signatures.

·

Basic reporting

The article is well written, and I have little trouble understanding the writing. The author provides clear literature citation and background where relevant, and frequently link their results with results of past studies.

The figures are the ones that I have trouble reading, because they are too small and have too low a resolution for them to be properly read. Figure 2B and Figure 8A and E are prime examples. The texts there are just not readable. For Figure 5, the texts can be read if more efforts are put in, but enlarging the font or splitting them into two figures may help readability. Figure 3B also has unreadable texts to the left of each line.

Experimental design

The experiment and analysis is well designed and described. The methods described in the materials and methods are in sufficient details for replication.

Validity of the findings

The main question asked by the study is whether a glycolysis-related gene pairs (GRGPs) can be used to predict prognosis of HCC, and the answer is yes. The authors find that seven GRGPs can be used as molecular biomarkers to predict HCC. Authors also find that the low-risk group based on these seven GRGPs have better overall survival (OS). Overall I find that the authors have strong data and analysis to back up their conclusions.

I only have a few comments to improve the manuscript:
1. Line 155: authors say that glycolysis ranked first. It is not immediately apparent in Figure 1 how glycoloysis rank first. It will be appreciated if authors can provide clearer explanation.
2. Figure 5E and 5F: more explanation is needed to explain how to interpret the heat map. There are only two colors, red and green, in the heat map, and they supposedly correspond to high and low levels, whereas there is another color scheme for high and low risk. I am confused by this heat map.
3. Line 290: CTMN1 is not in the seven GRGP and not in Table 1. It suddenly appears in the discussion here.
4. Figure 3: it is said that the vertical dashed line is at the optimal log (lambda) value. There are two vertical lines. Which one?

Additional comments

The writing of the article is excellent. There are just a few corrections that the authors may have missed:
1. Line 127: 'Data about 243 patients...'. Please change to 'Data of about 243 patients...'
2. Line 143: 'considered statistical significance' to 'considered statistically significant'
3. Line 203: 'largest contribution' to 'largest contributor'
4. Line 283: 'poor prognostic' to 'poor prognosis'
5. Line 297: ' which related with...' to 'which is related to...'

Reviewer 3 ·

Basic reporting

1. The writing has a significant number of grammatical and phrasing errors, wherein sentences are sometimes confusing. The writing needs significant improvement.
2. In some places, the GRGPs is misspelled as GPGRs. I believe this is a typo.
3. Most figures are extremely poor quality currently. Multiple figures such as Figure 2 have text that is too small to read. While, other figures have texts of different sizes, while others such as Figure 1 have words overlaid onto the figure itself, making it hard to see. Many figures seem like they were taken as programmed output and directly put into the final figure, this is unacceptable.
4. Figure legends have to be significantly improved and need to be more clear. Such as in Figure 4A, lower panel it is unclear what the numbers indicate.

Experimental design

No comment.

Validity of the findings

No comment.

---

## Round 0.2 · Minor Revisions

Please address the remaining issues pointed out by the reviewers and revise your manuscript accordingly.

·

Basic reporting

The author addressed the questions raised in the review comments. And they also made effort to improve the reliability of the model by generally comparing more glycolysis-related mRNAs in this new version of manuscript, rather than just comparing 26 DEmRNAs. And the conclusion makes more sense than the previous version.
However, there are some parts that doesn’t read smooth in gramma:
1. PDF file, material and methods section, line 95-96: ‘To explore whether the specific gene sets were significant differences between tumor group and normal group’, should be ‘significantly different’
2. PDF file, Line 129, ‘were used to verify whether the GRGPs signature was significant associated with OS’ should be ‘significantly associated’

Another thing is about the figure, Fig 2C doesn’t have descriptions in the figure caption.

Experimental design

The experiment design was reasonable. As mentioned above, instead of comparing 26 DEmRNAs, more RNAs are considered and more filters are used to indicate the relavent RNAs. This is a better design than the previous version.

Validity of the findings

The TCGA datasets were training sets, and the other datasets were used as validation datasets. From the report, the tests on the validation datasets fits the model well.

·

Basic reporting

The manuscript is written relatively clearly. The figure sizes and text readability have significantly improved from the first manuscript. Explanations to each of the figure have also improved, even though I find that there are more explanations needed for some figures.

In general, the manuscript seems to be written for a very specialized audience, assuming that the audience understands the field very well. A lot of explanations for interpreting graph, etc, are missing, so it is hard for people outside the field to understand the manuscript. I suggest that the authors take some time to provide more explanations to certain concepts, e.g. the lasso model, ROC, nomogram, etc.

There is one grammatical correction I hope to suggest:
- Line 47, 57, 224, etc. "have been proved" can be changed to "have been shown"

Here are some of the concepts that can benefit from more explanations:
- Figure 2A and 2B. What is the purpose of showing the graph? What should be our takeaway? The text says that lasso regression model revealed that 17 GRGPs as independent prognostic factors. From Figure 2A, should I focus on the lowest partial likelihood deviance? What is partial likelihood deviance? What is lambda? What are the numbers on top of Figure 2A? What do the two vertical lines represent? These are not explained. For Figure 2B, what does each colored line represent?
- For Figure 2C, there is no explanation of it in the Figure itself.
- Figure 6A and E. How to read a nomogram? Do we just draw a vertical line? My understanding is the line can be at a certain angle, not necessarily vertical. If you can provide an example, say a person with risk score of -1 and at stage III, what will be the 1 year survival? What do 'points' and 'total points' mean, and how are they derived?
- Figure 6D and H, it helps for readers to understand the graph if you explain that higher area under the curve (AUC) means better modeling
- For Figure 6H, the five year survival is almost 0.5. Can you comment on that?

Experimental design

In general, the study is well designed.

Validity of the findings

I find the findings to be valid, and it helps that the authors discusses the limitation of the study, in addition to its potential as biomarkers and targets for patients with HCC.

Reviewer 3 ·

Basic reporting

No Comment

Experimental design

No comment

Validity of the findings

No comment

Additional comments

I think the authors have sufficiently addressed all the concerns, and this manuscript looks good for publication now.

---

## Round 0.3 · accepted · Accept

Thank you for addressing the remaining issues pointed by the reviewers and for the adequate revision of the manuscript. I am glad to let you know that the revised version is acceptable now.